# YUCCA-Mediated Biosynthesis of the Auxin IAA Is Required during the Somatic Embryogenic Induction Process in *Coffea canephora*

**DOI:** 10.3390/ijms21134751

**Published:** 2020-07-03

**Authors:** Miguel A. Uc-Chuc, Cleyre Pérez-Hernández, Rosa M. Galaz-Ávalos, Ligia Brito-Argaez, Víctor Aguilar-Hernández, Víctor M. Loyola-Vargas

**Affiliations:** 1Unidad de Bioquímica y Biología Molecular de Plantas, Centro de Investigación Científica de Yucatán, A.C. Calle 43 No. 130 × 32 y 344 Col. Chuburná de Hidalgo, Mérida C.P. 97205, Mexico; miguel.uc@cicy.mx (M.A.U.-C.); cleyre_aph@hotmail.com (C.P.-H.); gaar@cicy.mx (R.M.G.-Á.); lbrito@cicy.mx (L.B.-A.); 2Catedrático CONACYT, Unidad de Bioquímica y Biología Molecular de Plantas, Centro de Investigación Científica de Yucatán, Mérida C.P. 97205, Mexico; victor.aguilar@cicy.mx

**Keywords:** auxin, *Coffea canephora*, localization, somatic embryogenesis, YUCCA, yucasin

## Abstract

Despite the existence of considerable research on somatic embryogenesis (SE), the molecular mechanism that regulates the biosynthesis of auxins during the SE induction process remains unknown. Indole-3-acetic acid (IAA) is an auxin that is synthesized in plants through five pathways. The biosynthetic pathway most frequently used in this synthesis is the conversion of tryptophan to indol-3-pyruvic acid (IPA) by tryptophan aminotransferase of *Arabidopsis* (TAA) followed by the conversion of IPA to IAA by enzymes encoded by *YUCCA* (*YUC*) genes of the flavin monooxygenase family; however, it is unclear whether YUC-mediated IAA biosynthesis is involved in SE induction. In this study, we report that the increase of IAA observed during SE pre-treatment (plants in MS medium supplemented with 1-naphthaleneacetic acid (NAA) 0.54 µM and kinetin (Kin) 2.32 µM for 14 days) was due to its de novo biosynthesis. By qRT-PCR, we demonstrated that *YUC* gene expression was consistent with the free IAA signal found in the explants during the induction of SE. In addition, the use of yucasin to inhibit the activity of YUC enzymes reduced the signal of free IAA in the leaf explants and dramatically decreased the induction of SE. The exogenous addition of IAA restored the SE process in explants treated with yucasin. Our findings suggest that the biosynthesis and localization of IAA play an essential role during the induction process of SE in *Coffea canephora*.

## 1. Introduction

Plants, unlike animals, have a high capacity for regeneration of new individuals identical to the mother from a cell or groups of cells without the need for fertilization. This regeneration mechanism is known as somatic embryogenesis (SE) [1,2]. SE is the development of structures similar to a zygotic embryo from somatic cells [3,4]. It can also be the process by which somatic cells, under induction conditions, generate competent cells that undergo a series of morphological, biochemical, and molecular changes to give rise to somatic embryos without the fusion of gametes [5]. SE provides an invaluable tool for the genetic improvement of plant species that cannot be propagated sexually [6].

The study of the biochemical and molecular mechanisms of SE allows us to identify the factors involved in SE induction [6] and to determine how best to apply them to the genetic improvement of a range of plant species [6,7]. Furthermore, SE is an example of totipotency because the somatic cells respond directly to a stimulus, leading to the development and formation of the somatic embryo. Therefore, SE is an excellent system for the study of cellular differentiation and dedifferentiation [8].

SE is a complex process that involves many factors, including plant species, tissue type (explant), culture medium, exogenous and endogenous growth regulators, and nitrogen and carbon sources [2,9,10]. In addition, somatic cells can activate the genetic machinery necessary for the transcription of genes involved in SE induction [11], implicating the alteration of cell wall composition and changes in growth regulators, gene expression, and epigenetic regulations in this process [12].

It has been proposed that plant growth regulators, mainly indole-3-acetic acid (IAA), play a crucial role in mediating the signal transduction that leads to the reprogramming of gene expression. This change is followed by a series of cell divisions that induce disorganized growth (callus) or lead directly to SE [13]. IAA is a molecule that controls almost all aspects of plant growth and development [14]. Its biosynthesis is crucial for plant homeostasis, including embryo development, fruit ripening, organogenesis, and plant architecture [15,16]. However, the action of auxin is determined by its synthesis and distribution in tissue, mainly via its polar transport from cell to cell [17,18].

The route most conserved and providing the most direct way to produce IAA in plants is from tryptophan via two enzymatic reactions consisting of tryptophan aminotransferase of *Arabidopsis* (TAA) and YUCCA (YUC) flavin monooxygenase of the indole-3-pyruvic acid (IPA) pathway [14,19,20,21].

Genetic studies have demonstrated that YUC functions as the rate-limiting step of the IPA pathway, indicating that YUC plays a crucial role in developmental processes regulated by cellular IAA levels [14]. Biochemical and molecular studies have shown that these gene families (*TAA* and *YUC*) participate in the pathway of IAA biosynthesis in several plant species, including *Arabidopsis thaliana*, *Zea mays*, and *Oryza sativa* [22,23]. It has been reported that IAA biosynthesis through *YUC* is necessary for the establishment of the basal part of the embryo and onset of embryonic organs [24]. Previous findings indicated that the location of auxin biosynthesis plays an essential role in many growth and development processes, including embryogenesis [25].

Cheng et al. [26] overexpressed *YUC1*, *YUC2*, *YUC4*, and *YUC6* genes, and their results indicated an increase in the production of auxin in *Arabidopsis* seedlings [26]. In addition, they determined the expression of *YUC1* and *YUC4* at the apical meristem and primordia of young leaves [24,26]. Single or double mutants showed no adverse effect, unlike quadruple mutants, which showed severe effects in the development of the seedlings [24]. Accordingly, due to redundant functions of the *YUC* genes family it is difficult to access reverse genetic approaches to understand the physiological role of IAA biosynthesis [14]. Hence, the use of specific inhibitors to overcome the redundant activity of target genes has emerged as a useful tool for genetic studies [14].

Despite the various studies in this area, the genes regulating IAA auxin biosynthesis during embryogenesis are not known [24], and endogenous intracellular levels remain unclear during the SE induction process. Apparently, de novo IAA biosynthesis plays an essential role in SE, because previous reports have shown that auxin biosynthesis is dynamic during embryogenesis [27].

Our primary goal in this work was to determine whether YUC-mediated IAA biosynthesis is involved in the SE induction process in *Coffea canephora*. To answer this question, we used RT-qPCR to measure the transcription levels of *CcYUC* and used a specific yucasin inhibitor to block the biosynthesis of the auxin IAA. Yucasin is a powerful specific YUC enzyme inhibitor [28].

In this study, we found that *CcYUC1*, *CcYUC1*-*putative*, *CcYUC4*, and *CcYUC*-*like* have dynamic expression patterns at the moment of the induction of the SE process. We showed that there exists a correlation between the *CcYUC* expression pattern and the location of the free IAA auxin signal at the beginning of the induction of the SE process. Furthermore, the formation of a local endogenous IAA gradient in specific tissues was crucial during the SE induction process in *C. canephora*. On the other hand, treatment with yucasin inhibited SE, but exogenous IAA addition restored the embryogenic process. Our data showed that the YUC-mediated IAA biosynthesis is crucial for SE in *C. canephora*.

## 2. Results

### 2.1. The Induction Process, Histology, and Expression Profiling of the CcYUCs Transcribed during SE in C. canephora

The pre-treatment stage (plants in MS medium supplemented with NAA 0.54 µM and Kin 2.32 µM for 14 days) and growth regulators are essential for SE in *C. canephora*. To induce SE, we used foliar explants of plantlets maintained in Murashige and Skoog (MS) medium for 14 days under photoperiod conditions (see Materials and Methods). The explants were then transferred to an auxin-free medium supplemented with benzyladenine (BA, 5 μM). The samples were collected 56 days after induction (dai) (Figure 1A) and the number of somatic embryos was counted (Figure 1B). During the induction process, we observed a rapid proliferation of proembryogenic cell mass at the edge of the explant wound 14, 21, and 28 dai. By 56 dai, all the developmental stages of the somatic embryos were found (G, globular; H, heart; T, torpedo; and C, cotyledonary; Figure 1A). The total embryo production per flask was 402.3. A total of 298 were embryos in the G stage, followed by 48.6 embryos in the H stage, 31.3 in the T stage, and 24.3 in the C stage (Figure 1B).

Transversal cuts of the explants were analyzed during the SE induction process, in order to observe the changes that are carried out in the explant and the formation of the first embryogenic structures. The results showed that at the beginning, the explant tissues were composed of spongy and palisade mesophyll cells (Figure 2A). The structure of the explants showed almost no change during the first 14 days of the induction of SE (Figure 2B,C). After 21 days in the induction medium, the first embryogenic cells appeared. These first structures were located near the vascular tissue (Figure 2D). These new cells were small, circular, and had a very dense cytoplasm (Figure 2E). Twenty-eight dai, there was an increase in the proembryogenic mass, with most of the mass emerging from spongy mesophyll cells (Figure 2F). The formation of proembryos is the result of the coordinated growth of a series of organized cell divisions that will give rise to somatic embryos.

### 2.2. Identification and Content of Free IAA and Conjugated IAA

IAA is found in cells in free and conjugated form. In all the systems in which IAA conjugates have been measured, they are more than 90% of the total auxin. We used liquid chromatography–mass spectrometry (LC-MS/MS) and compared the retention times and fractionation patterns of standards and the auxins extracted from the explants. In this way, we identified free IAA and its conjugates with aspartic acid (Asp), glutamic acid (Glu), alanine (Ala), and leucine (Leu). The elution and fractionation patterns correspond perfectly between the standard and the samples (Appendix A). With this certainty, we proceeded to quantify the free IAA and its conjugates. The amount of IAA-Asp determined was only at the trace level, so the quantification was performed in the other three conjugates and free IAA (Figure 3).

The endogenous initial free IAA content was 0.22 nmol g^−1^ FW and increased more than nine times during the pre-treatment of the seedlings in the presence of NAA and Kin (Figure 3), and reached a maximum content of 2.06 nmol g^−1^ FW fourteen days after the start of the pre-treatment. The explants were taken from these plantlets to start the induction of SE. The free IAA content decreased rapidly during the first hour of explant incubation in the induction medium of the SE. and was maintained at levels of 0.1 to 0.5 nmol g^−1^ FW for the next six days. Its level increased again with the appearance of the first embryonic structures.

IAA conjugates are a significant part of IAA homeostasis [29,30], so they were assessed throughout the entire process (Figure 3). The conjugate with glutamic acid made up more than 85% of the total IAA content. Fourteen days of pre-treatment produced an endogenous level of 98 nmol g^−1^ FW of IAA-Glu. After the induction of SE, the IAA-Glu content decreased seven times in just 24 h and practically disappeared seven days after the induction of SE.

The conjugates with alanine and leucine made up 12.6% of the total IAA. These conjugates are very important for homeostasis of IAA, since they can be hydrolyzed and contribute to free IAA. The IAA-Ala increased from 2.49 nmol g^−1^ FW at the beginning of the pre-treatment to 11 nmol g^−1^ FW at the time of induction and decreased very quickly over the next 21 days. The IAA-Leu conjugate decreased during the first seven days of pre-treatment, and by the time of the induction of SE, had returned to its initial levels. This conjugate decreased very rapidly during the first hours of the induction of SE and then began to increase to levels ranging from 2.5 to 5.5 nmol g^−1^ FW in the following days (Figure 3).

To determine whether the increase in IAA content observed during pre-treatment of *C. canephora* seedlings was due to de novo biosynthesis, 3-^14^C-Trp was used, as has been done in *A. thaliana* [31,32,33] and *Solanum lycopersicum* [34].

First, we performed standard thin-plate chromatography, and ran real non-radioactive samples and identified the compounds by LC-MS. This experiment gave us the confidence to associate radioactive spots with their identity. The result of incubation in the presence of 3-^14^C-tryptophan (Trp) can be seen in Table 1. As the days of the pre-treatment progressed, there was an increase in the radioactivity associated with the IAA. This result suggests that the IAA biosynthesis was de novo from Trp. A 7-fold increase in the IAA content was observed between Day-9 and Day 0 (Table 1). The marked Trp only began to accumulate on Days-4 and 0, possibly because the cells had reached a maximum biosynthesis of IAA.

In an experiment using a radioactive label, it is important to determine the fate of the entire radioactive label. Therefore, we measured the radioactivity present in each of the fractions of the experiment. A part of the radioactivity was in the tissues and other part in the culture medium. In the case of tissues, radioactivity was not only determined in the explants used to induce SE, but also in the rest of the plant. The extraction of auxins present in the stem of the plants was also performed and their radioactivity was measured. From Days −9, −7, and −4, the stem was not extracted because it was submerged in the liquid medium and it would not have been possible to determine how much radioactivity was due to the auxin present in the stem and how much radioactivity was external pollution; however, this determination was done on Day 0. The seedlings were incubated in 10 mL of pre-treatment medium to which the radioactivity was added; of this, 10 µL was taken to count in the scintillator. A total 2,679,807 cpm was added to each experimental unit (Table 1). As can be seen in Table 1, the destination of most of the radioactivity used in each experimental unit was determined.

SE is a complex process that involves multiple factors, including the biosynthesis of IAA through the TAA/YUC pathway [35]. To test whether the *CcYUC* genes were participating during the SE induction process in *C. canephora*, we performed a quantitative expression analysis of *CcYUC* transcript levels. First, we determined how many *YUC* genes were in the genome of *C. canephora* (http://coffee-genome.org/ [36]). The search yielded eight *YUC* genes. Two copies of the *CcYUC1* gene and two copies of the *CcYUC10* gene were found throughout the genome. At the same time, we performed an analysis of the transcriptome of *C. canephora* [37]. The results showed the presence of five transcripts of *CcYUC* gene products during the induction of SE. Two of these five transcripts belonged to the *CcYUC1*. The locus of the *CcYUC1* gene is on Chromosome six in the genome of *C. canephora*, and structurally consists of five exons and four introns with a length of 3060 base pairs (Figure 4A). The reading frame of the *CcYUC1* gene predicted a 519 amino acid protein of 59 kDa molecular mass, with FAD motifs (Figure 4B) characteristic of flavin monooxygenase enzymes. To gather a complete picture of the participation of the *CcYUC* family during the induction of SE, we analyzed the expression of seven *CcYUC* genes, including the two copies of the *CcYUC1* gene, *CcYUC3*, *CcYUC4*, *CcYUC6*, and *CcYUC10* genes and one *CcYUC*-*like* gene (Appendix A).

The analysis of the *CcYUC* expression was performed by qRT-PCR for samples taken on the −14, 0, 7, 14, 21, and 28 dai of SE. Among the *CcYUC* genes analyzed, it was found that the *CcYUC1* gene was especially highly expressed, and its transcription level showed a distinguishably substantial increase on the zero days of SE induction (up to 6.8-fold) (Figure 5A). On Days 7, 14, and 21 of SE, the transcriptional activity of *CcYUC1* decreased but then had a slight increase on Day 28 (up to 1.2-fold). In contrast, *CcYUC1*-*putative* was only expressed on Day 0 (Figure 5B), and *CcYUC4* had two expression peaks, on Day 0 (up to 4.2-fold) and at 28 dai (up to 2.7-fold), respectively (Figure 5C). *CcYUC3* (Figure 5D) and *CcYUC6* (Figure 5E) genes showed low levels of transcripts compared to Day −14 throughout the entire SE induction process. CcYUC10 had very similar expression times 7 and 21 dai of SE (Figure 5F). The *CcYUC*-*like* gene was expressed only during the pre-treatment stage, with a 3-fold increase at the end of the pre-treatment and just before the explants were introduced into the induction medium (Figure 5G). Once in the explant was in the presence of the induction medium, the expression of the *CcYUC*-*like* gene disappeared. Of the seven analyzed *CcYUC* genes, four genes, including *CcYUC1*, *CcYUC1*-*putative*, *CcYUC4*, and *CcYUC*-*like*, were upregulated on Day 0, while *CcYUC3* and *CcYUC6* were downregulated during the SE induction process. The behavior of *CcYUC10* did not follow a definite pattern along with the stages of induction of SE.

### 2.3. Endogenous Free IAA Accumulation and Localization during the SE Induction Process

To investigate whether there was a specific localization of IAA in leaf explants of *C. canephora* during the process of SE induction, we used an anti-IAA mouse monoclonal primary antibody specific for free IAA and an Alexa Fluor 488-labeled anti-mouse IgG expand secondary antibody.

Previously, it was reported that auxin response gradients were established in specific regions of the embryonic callus and were responsible for SE [38,39].

Cross-sections of leaf tissue, 30 µM thick, were made of explants of *C. canephora* leaf during the process of induction of SE. After pre-treatment, we observed the cells that were part of the tissue structure. This tissue was made of spongy and palisade mesophyll cells (Figure 6A,G). In addition, although no morphological change was observed between the control explants and those treated with 100 µM yucasin, there was a difference in the thickness of the explant. In tissues treated with 100 µM yucasin, it was observed that the thickness of the tissue was thinner compared to the control (Figure 6A,G). It is possible that the yucasin inhibitor caused this effect.

In our study, on Day 0, we found a strong free IAA signal during the SE induction process (Figure 6B). The signal decreased from 7 dai (Figure 6C) through 14 dai (Figure 6D). At 21 dai (Figure 6E), the IAA signal began to increase. Seven days later, the increase of the IAA signal was much more significant (Figure 6E), and was localized at the edges of the explants and in the cell walls of the spongy mesophyll cells (Figure 6E).

On the other hand, the immunolocalization assays of free IAA of the yucasin-treated samples revealed important changes in the IAA signal accumulation pattern. At Day 0 of the SE induction process, we found a free IAA auxin signal in explants treated with 100 µM yucasin (Figure 6H). The free IAA signal found was less intense than in the control samples (Figure 6B). The IAA signal disappeared from Day 7 to Day 21 after the induction of SE (Figure 6I–K). An essential difference from the control samples was the decrease in the IAA signal, in the presence of the yucasin, at 28 dai (Figure 6L) compared with the control at the same stage (Figure 6F).

The next step was to determine, intracellularly, the location of the IAA signal (Figure 7). The endogenous accumulation of free IAA was located in the interior chloroplasts and nucleoplasm of spongy mesophyll cells (Figure 7B).

In control tissues, numerous chloroplasts were located within the spongy mesophyll cells (Figure 7A). These cells had prominent nuclei (Figure 7B). The free IAA signal was found both in the chloroplasts and in the nuclei of the spongy mesophyll cells (Figure 7E). We showed that IAA free of mouse anti-IAA monoclonal primary antibodies was specific (Figure 7D) and the signal was not mixed with chlorophyll (Figure 7C). In treatments with 100 µM of yucasin, the free IAA signal was found in the cytosol (Figure 7I) and not in the chloroplasts or nuclei (Figure 7G–J), as in the control.

### 2.4. Effect of the Inhibition of Auxin IAA Biosynthesis by Yucasin during the SE Induction Process in C. canephora

Several studies have shown that the pathway of IAA biosynthesis in most plants occurs through two simple steps from tryptophan, mediated by TAA and YUCs [19,20,40]. The tryptophan-dependent pathway for IAA biosynthesis through *YUC* genes could be a determining factor for the development of embryogenesis [15]. At the same time, different groups have shown that an increase in the amount of IAA is required to initiate the induction of SE [41,42]. Our results (Figure 3) showed an increase in the IAA content during the pre-treatment phase. To test whether *CcYUC*-mediated IAA biosynthesis was required for the SE induction process, we used the auxin biosynthesis inhibitor, yucasin, which specifically inhibits the function of YUC proteins [28].

The results showed that the efficiency in the formation of proembryogenic mass was severely affected by the treatment in a dose-dependent way (Figure 8A). Twenty-one dai, the explants treated with the inhibitor showed a decrease in proembryogenic mass formation, particularly at 20, 50, and 100 µM of yucasin (Figure 8A). After 56 days of SE, all of the explants from plants incubated in the presence of yucasin during pre-treatment showed signs of damage, including tissue necrotization and phenolization (Figure 8A). The treatments with 50 and 100 µM of yucasin completely inhibited the development of the proembryogenic mass from 28 dai on (Figure 8B).

The presence of free IAA decreased significantly after the explants were transferred to the SE induction medium (Figure 3). However, this small amount of IAA was very important for the induction of the embryogenic process. When yucasin was added, this small amount disappeared and the embryogenic process did not take place (Figure 9).

Quantification of the number of embryos produced 56 dai showed a significant decrease in the number of embryos produced by the explants exposed to the yucasin inhibitor (Figure 10). In the presence of the inhibitor, only globular-shape embryos were formed. The decrease varied from 72 to 94% of the control. Even the lower concentration of the inhibitor produced a sharp decrease in the number of embryos. In the presence of 10 µM–100 µM of yucasin, the number of globular embryos was less than 40 embryos per flask, in comparison with 300 globular embryos per flask for the control.

### 2.5. Restoration of Somatic Embryogenesis by Exogenous Addition of IAA

To confirm that the effect seen on somatic embryo production was due to the inhibition of IAA biosynthesis as a result of yucasin treatment, we added 1.0 µM of IAA to the medium of induction of the SE containing the explants treated previously with 100 µM yucasin-induction medium. Twenty-eight days after the exogenous addition of IAA, the embryogenic process inhibited by the yucasin was restored (Figure 11A). It was possible to see all of the stages of development after the addition of exogenous IAA (Figure 11B,C). In contrast, the samples treated with yucasin but not exogenous IAA did not produce embryos, beyond the few globular embryos already present at the beginning of the experiment (Figure 11D,E). The somatic embryos produced after the addition of the exogenous IAA were entirely normal, and the somatic embryos reached the cotyledonary stage (Figure 11F). The production of somatic embryos in the induction medium containing yucasin + exogenous IAA was more than 77% higher than in the control without yucasin (Figure 11G).

## 3. Discussion

Several biochemical and genetic studies of SE have been reported, involving *A. thaliana* [43], *Brassica napus* [44], *Medicago truncatula* [45], *Coffea* spp. [2,46], and many other species [4,47,48]. However, the mechanism by which somatic cells change their genetic program and become somatic embryos is not yet fully understood.

SE is a complex process and is strictly regulated. In this work, we focused on IAA biosynthesis mediated by YUCs. It has been reported that auxin transport [17,49] and signaling plays an essential role throughout the entire life cycle of plants, including embryogenesis [16,50,51,52].

In this study, we showed that *CcYUC*-mediated IAA biosynthesis is required during the SE induction process in *C. canephora*. Histological analysis showed that at the beginning of the proliferation of the proembryogenic mass 21 dai, the embryogenic cells appeared near the vascular tissue (Figure 2D). The formation of embryogenic structures was observed from 28 dai onwards. By 56 dai, all of the different development stages of the somatic embryos of *C. canephora* were present (Figure 1). These data were in line with the well-documented fact that mesophyll cells located near vascular bundles of leaves are the first to divide [53,54]. These cells produce up to fivefold more proteins than non-embryogenic cells [55]. The embryogenic cells are characterized as small, isodiametric, and densely cytoplasmic. These cells then undergo a series of successive divisions to give rise to a somatic embryo [11].

The increase in the level of IAA during the pre-treatment period [41] (Figure 3) could have been due to de novo biosynthesis. The increases in the expression of *CcYUC1*, *CcYUC1*-*putative*, *CcYUC4*, and *CcYUC*-*like* during the pre-treatment supported this assumption (Figure 5). Consistent with this hypothesis, qRT-PCR expression analysis of the CcYUC genes during the SE induction process showed that most (5/8) of the CcYUC genes encoded in the genome were transcriptionally active at the beginning of the SE induction. The transcription levels on Day 0 were congruent with the free IAA signal found in the explants in the induction medium. The *YUC* gene family encodes flavin monooxygenase enzymes for the biosynthesis of IAA from IPA [20]. Biochemical and genetic studies indicate that plants use Trp as a substrate, which is converted to IPA as an intermediary for the production of IAA [21]. The use of 3-^14^C-Trp permitted us to confirm that the increase in the IAA content came from de novo biosynthesis (Table 1).

In situ hybridization has determined YUC1 and YUC4 expression at the apical meristem and primordia of young leaves during organogenesis in *A. thaliana* [26,56,57]. These genes are activated quickly after wounding (within 4 h), suggesting that these two genes participate in the production of endogenous auxin in leaf explants [57]. In this same model, the overexpression of *YUC* genes increased the endogenous content of IAA in young leaves. The mutation of the *YUC* genes produced a drastic decrease in the content of IAA [21,58,59]. Similarly, in maize, YUC mutations have been shown to cause a reduction of IAA levels and disturb the vegetative development [60].

In *A. thaliana*, the inhibition of *YUC* genes prevents the expression of WOX11, resulting in the blocking of rooting [57]. This labor is divided among the different *YUC* genes. *YUC1* and *YUC4* are expressed suddenly in response to wounding after detachment in both light and dark conditions and promote auxin biogenesis in both mesophyll and competent cells [57]. These two genes are also expressed at the beginning of the induction of the SE in *C. canephora* (Figure 5), suggesting a similar role in both species. However, the inactivation of a single YUC gene does not cause developmental defects, due to the redundant function between *YUC* genes in *A. thaliana* [24,26]. Therefore, we used a specific inhibitor to block the function of YUC enzymes in *C. canephora*. Yucasin inhibits the activity of YUC enzymes and suppresses the effect of the high-auxin phenotype of YUC overexpression found in *A. thaliana* [14]. The results shown in Figure 8 revealed that the inhibition of IAA biosynthesis (Figure 9) strongly affected the progress of SE in *C. canephora*. The immunolocalization of IAA in *C. canephora* explants treated with the inhibitor (Appendix A) shows no signal in the explants exposed to yucasin (Appendix A).

The exogenous addition of IAA to explants treated with yucasin restored the SE process (Figure 10). These results revealed that the YUC-mediated biosynthesis of the auxin IAA is critical for SE in *C. canephora*. The IPA pathway is highly conserved in land plants; however, since IAA can be synthesized through five different routes, one of them independent of tryptophan (from indole-3-glycerol phosphate [61,62]), a contribution to the IAA pool from some combination of the other four routes cannot be rule out.

In this study, we used different concentrations of yucasin (5, 10, 20, 50, and 100 µM). We showed that yucasin inhibited the production of somatic embryos in explants of *C. canephora* (Figure 10). The action of IAA occurs in its free form and acts in the nucleus on the expression of auxin-response genes [63]. Different concentrations of this auxin may give rise to various physiological processes [64]. Its synthesis and distribution in tissue determine the action of auxin, mainly via its polar transport during SE in *A. thaliana* [17,65]. During zygotic embryogenesis, IAA is regulated by its biosynthesis and spatiotemporal localization through specific carriers of auxin PIN (pin-formed) and ABCB (ATP-binding cassette protein subfamily B) [66,67,68].

On the other hand, the IAA found in treatments with 100 µM yucasin was possibly due to two factors: first, the release of IAA through hydrolysis of IAA conjugates. Conjugation plays a central role in the homeostasis of IAA [58,69], and this reaction is catalyzed by the Gretchen Hagen 3 enzymes (GH3) family of acyl acid-amido synthetases [70]. The IAA found in this work in explants with yucasin could have been the result of IAA-Leu-resistant (IRL) enzyme activity. In the absence of de novo biosynthesis of IAA (by the inhibition of yucasin), the IAA conjugates hydrolyze to leave it in its free form [71]. IAA metabolism depends on which amino acid is attached; for example, the conjugation of IAA with alanine or leucine results in a form that is stored but can be easily hydrolyzed [72]. Auxin conjugates are hydrolyzed to release IAA to maintain intracellular homeostasis in tissues in response to environmental conditions [73]. Recently, hydrolysis of aspartic (IAA-asp) and glutamic (IAA-glu) conjugates was reported in strawberry plants to provide free IAA for fruit growth [74]. The second factor that could explain the presence of IAA in explants treated with yucasin is the existence of an alternate route to produce IAA [58]. Indole-3-acetaldehyde (IAAld) has been proposed as an intermediary in the IAA biosynthesis pathway, since some bacteria produce IAA from IPA using IAAld as an intermediary [75]. However, there is not enough evidence for this theory. Therefore, IAAld is unlikely to participate in the IPA pathway [21]. Another auxiliary route in the biosynthesis of IAA is from indole-3-acetaldoxime (IAOX) [31]. CYP79B2 and CYP79B3 catalyze the conversion of Trp to IAOx [76]. However, it has been reported that the IAOx pathway is specific to Brassicaceae plants, because CYP79B genes are very limited in these species [31].

However, many questions remain unanswered about the de novo biosynthesis of IAA during the SE induction process. Several factors are implicated in its induction, including alteration of the cell wall composition, changes in growth regulators, genetic expression, and epigenetic regulations [12]. Furthermore, although it is believed that the predominant route in auxin synthesis is from IPA, the molecular mechanisms that regulate biosynthesis at the transcriptional level and protein level are unknown [77].

In summary, the data in this research suggest that the pre-treatment of the coffee plantlets produced an increase in the level of IAA. This increase was due to de novo biosynthesis, and the presence of IAA at the beginning of the induction of SE in *C. canaphora* is indispensable for the process to begin.

## 4. Materials and Methods

### 4.1. Plant Material and Growth Conditions

*C. canephora* plantlets were propagated and maintained under in vitro photoperiod conditions 16/8 h (150 µmol m^−2^ s^−1^) at 25 ± 2 °C in MS inorganic culture medium ([78], PhytoTechnology Laboratories, M524, Shawnee Mission, KS, USA). The MS medium contained 29.6 µM thiamine-HCl (Sigma, T3902, St. Louis, MO, USA), 550 µM myo-inositol (Sigma, I5125, St. Louis, MO, USA), 0.15 µM L-cysteine hydrochloride hydrate (Sigma, C8277, St. Louis, MO, USA), 16.24 µM nicotinic acid (Sigma, N4126, St. Louis, MO, USA), 87.64 mM sucrose, and 0.25% (*w*/*v*) CultureGel TM Type I-Bio Tech Grade (PhytoTechnology Laboratories, G434, Shawnee Mission, KS, USA), pH 5.8. Plantlets were subcultured every 6 weeks by in vitro transplantation of shoot intermodal segments to fresh maintenance media.

### 4.2. Induction of Somatic Embryogenesis in C. canephora

For the induction of SE, we started with 4 month old plantlets of *C. canephora* cultured under in vitro conditions. A batch of plantlets was selected and placed in semisolid medium for pre-treatment. The culture medium was MS medium, supplemented with 0.54 µM NAA (Sigma N1145, St. Louis, MO, USA) and 2.32 µM Kin (Sigma K0753, St. Louis, MO, USA), for fourteen days under photoperiod conditions (16 h light/8 h dark) at 25 ± 2 °C. For the induction of SE, Leaves two and three were used. The explants were cut into circles of approximately 0.25 cm in diameter and transferred to Yasuda liquid medium [79] supplemented with 5 µM BA (PhytoTechnology Laboratories, B800, Shawnee Mission, KS, USA). The cultures were incubated in the dark at 25 ± 2 °C with shaking (100 rpm) for 56 days [80]. Samples were taken 0, 7, 14, 21, and 28 days after induction (dai) of SE.

### 4.3. Extraction of Auxins and Their Conjugates

For the extraction of auxins and their conjugates, 100 mg of tissue was used from Days-14, -9, and -4 of pre-treatment; on Day 0 of the induction of SE and 0.02, 0.04, 1, 7, 14, and 21 days after the induction of SE. The samples were stored at −81 °C until use. The frozen tissue was ground with liquid nitrogen and mixed with 1 mL of acidic water (the pH was adjusted to 2.8 with HCl). The mixture was transferred to a test tube with an additional milliliter of acidic water. The mixture was stirred for 1 min with 1 mL of a solution of butylated hydroxytoluene (Acros Organics, 112992500, Thermo Fisher Scientific, NJ, USA), and then 1 mL of ethyl acetate (CTR Scientific 00184, Monterrey, Mexico) was added. The mixture was stirred for 1 min and the supernatant recovered. Next, 2 mL of ethyl acetate was added, stirred for 1 min, and the supernatant was recovered. This operation was repeated once more. From this mixture, 3 mL of the organic phase was taken and evaporated with nitrogen gas. The dried sample was resuspended in 1 mL of the mobile phase, filtered through a Millipore filter (0.22 µM) and analyzed using high-resolution liquid chromatography (HPLC) (60% acetonitrile; JT Baker 9017-03 (Thermo Fisher Scientific, NJ, USA): 40% water containing 0.5% (*v*/*v*) acetic acid; CTR Scientific, 00500, Monterrey, Mexico). The standards used were IAA (Sigma, I1250, St. Louis, MO, USA) and IAA-Ala (Sigma, 345911, St. Louis, MO, USA). Preparation of IAA-Glu, IAA-Leu, and IAA-Asp was previously reported [41].

### 4.4. High-Performance Liquid Chromatography

For the analysis of the samples, an Agilent Technologies 1200 high-resolution liquid chromatograph (HPLC) consisting of a quaternary array of pumps (Agilent Technologies G1311A, Santa Clara, CA, USA) connected to an automatic injector (Agilent Technologies G1329A, Santa Clara, CA, USA) was used. A total 20 μL of the tissue extract was injected and subjected to chromatography with an isocratic elution system with a flow rate of 0.6 mL min^−1^ in a C_18_ reverse-phase column (Phenomenex, Torrance, CA, USA) of 250 mm × 4.6 mm. The samples were analyzed with a fluorescence detector (Agilent Technologies G1321A, Santa Clara, CA, USA) at an emission length of 280 nm and an excitation length of 340 nm. The presence of compounds in the analyzed samples was determined by the retention times of IAA and of IAA-Ala, IAA-Leu, IAA-Glu, and IAA-Asp conjugates (Appendix A), for which co-injections of the standards and the samples were analyzed, to determine whether they co-eluted. The calibration curves were made with free IAA and the conjugate standards using the area under each curve for each compound.

### 4.5. Liquid Chromatography–Mass Spectrometry of Auxins

LC-MS/MS analysis was performed using a Thermo LTQ Orbitrap (Thermo Fisher, Carlsbad, CA, USA), equipped with a heated-electrospray ionization (HESI-II) source with sheath gas set to 60, auxiliary gas set to 20, source temperature set to 310 °C, and spray voltage 4 kV in a positive mode. To determine the chemical fragmentation of auxins, a solution of an individual auxin at a concentration of 100 µg mL^−1^ in methanol:water (80:20; *v*/*v*) was directly infused on LTQ Orbitrap at 5 µL min^−1^. The collision energy dissociation (CID) parameter for auxins was optimized to yield either parent-ion-dependent product ions (M+H)^+^ and nearly 20% of the parent ion. Chromatographic separations were performed using a reverse-phase ZORBAX Eclipse XDB C_18_ (150 × 4.6 mm i.d., 5 µm particle size, 80 Å pore size) column (Agilent Technologies, G1321A). A gradient of 0.1% formic acid in water (Solvent A) and 0.1% formic acid in acetonitrile (Solvent B) was used during LC separations. A flow rate of 0.3 mL min^−1^ was used, and the injection volume was 2 µL. The gradient program was 5% B at 0 min, to 20% B at 20 min, to 30% at 32 min, to 80% at 34 min, to 100% at 36 min, kept at 100% for 2 min, and then to 5% at 40 min and kept at 5% for 6 min. Retention time and spectra were processed with raw Xcalibur data files.

### 4.6. Preparation of Seedlings in the Presence of 3-^14^C-Trp

Seedlings were incubated in the presence of 3-^14^C-Trp (NEN-Dupond; 1.85 MBq 55 mC mmol^−1^) during the 14 days of pre-treatment in MS liquid medium supplemented with NAA 0.54 µM and Kin 2.32 µM. Auxins were isolated on Days −14, −9, −7, −4, and 0 of pre-conditioning. To monitor the incorporation of labeled Trp into the IAA, the auxinic extract was run on a silica TLC plate with an alumina fluorescent indicator Kieselgel 60 F254 (Merck, 105554, CDMX, Mexico). Five microliters of the leaf extract incubated with 3-^14^C-Trp and 5 µL of the Trp standards (0.25 µL; Sigma, T0254, St. Louis, MO, USA), indol-3-pyruvic acid (IPA, 1 µL; Sigma, L7017, St. Louis, MO, USA) and IAA (0.25 µL; Sigma 45533, St. Louis, MO, USA) was applied to the plates. The samples were run for 3 cm using a mixture of chloroform:ethyl acetate (50:50) as the mobile phase, and Salkowski reagent was used as a developer. Bands were identified by the R_f_ of the compounds. The silica of each band was scraped, deposited in vials with scintillation liquid, and the radioactivity of each was quantified in a scintillation counter (Beckman 6500, CDMX, Mexico).

To track the destiny of the all of the radioactivity used, we used the following protocol. Once the auxins were extracted from pre-treatment seedlings that were incubated in the presence of 3-^14^C-Trp, 5 µL of the total 100 µL of the leaf extract was placed on a chromatographic plate, as well as Trp, IPA, and IAA standards. The spots were developed with the Salkowski reagent (stain compounds containing an indole group). Each spot on the plate was associated with the corresponding standards. Each spot was scraped off the plate and placed in a scintillation vial to count the radioactivity present. In order to be certain that there were no radioactive compounds outside those marked by the developer, the areas between the spots were cut, and the radioactivity was determined. No radioactive label was detected in any case.

### 4.7. Yucasin Inhibition Assay

Yucasin, an inhibitor of the YUC protein function in the auxin biogenesis pathway (5-(4-chlorophenyl)-4H-1, 2, 4-triazole-3-thiol (Santa Cruz Biotechnology, 233161, Santa Cruz, CA, USA)), was added to the pre-treatment semisolid medium at concentrations of 5, 10, 20, 50, and 100 µM for fourteen days under dark conditions. Leaf explants of *C. canephora* plantlets treated with yucasin were transferred to the SE induction medium [5]. Yucasin was dissolved in dimethyl sulfoxide (DMSO, Sigma, D8418, St. Louis, MO, USA) and IAA was dissolved in EtOH (J. T. Baker, Thermo Fisher Scientific, Phillipsburg, NJ, USA). As a control, DMSO was added to the semisolid pre-treatment medium. The experiments were performed in biological triplicate. The effects of the different concentrations of the yucasin were analyzed by quantifying the number of embryos formed after 56 days.

### 4.8. Plant Tissue Sampling

The plant tissue samples were collected at different times from Days 0 (D0), 7 (D7), 14 (D14), 21 (D21), and 28 (D28). Samples collected 0, 7, 14, 21, and 28 dai of SE were used for performing immunolocalization assays. Days −14 (at the beginning of the pre-treatment), 0, 7, 14, 21, 28 dai were used for analysis of quantitative genetic expression.

### 4.9. Real-Time Quantitative Analysis of Gene Expression

The total RNA extraction was performed following the manufacture’s instructions for TRI reagent (Sigma, 93298, St. Louis, MO, USA). A total 100 mg of plant tissue was used for RNA extraction. The integrity and purity of the RNA was evaluated by 1% agarose electrophoresis and spectrophotometry (NanoDrop 2000, Thermo Scientific). Five milligrams of total RNA was used for cDNA (complementary DNA) synthesis using a Super Script II reverse transcriptase kit (Invitrogen) following the manufacturer’s protocol. Quantification of gene expressions by qRT-PCR were carried out with Appied Biosystems equipment using the Step One program. The Coffee “Genome” Hub page was consulted (http://coffee-genome.org/) and the genome database of *C. canephora* was downloaded for the design of specific primers described in Appendix A to analyze the YUCCA genes. With the support of the *C. canephora* transcriptome [37], an analysis was carried out to determine which genes could be the main participants during the induction process of SE. After identifying the YUC candidate genes, the coding sequences (CDS) of the specific genes were downloaded and the design of the primers was carried out in the Primer3plus program (http://www.bioinformatics.nl/cgi-bin/primer3plus/primer3plus.cgi/). The primers for each gene were tested via in situ PCR in Sol Genomics (http://solgenomics.net/).

### 4.10. Histological Analyses

The plant tissues were fixed in FAA solution (10% formaldehyde (Fischer BioReagents, BP531, Pittsburgh, PA, USA), 5% acetic acid (Sigma, 695092, St. Louis, MO, USA), and 50% ethanol (Meyer, 0390, CDMX, Meixo)) for 72 h in dark conditions at 4 °C. A gradient of sucrose (10, 20, 30%) was made to embed the samples in a PB buffer (10 mM sodium phosphate dibasic (Sigma, S3264, St. Louis, MO, USA) and 2 mM potassium phosphate monobasic (Sigma, P5655, St. Louis, MO, USA)), pH 7.2 (adjusted with NaOH 1 N), adding three to six drops of Leica tissue-freezing medium (Leica Biosystem, Code 14020108926, Guadalajara, Mexico) to the gradients of 20 and 30%, respectively. Each gradient was changed after 1 h at 4 °C. Subsequently, the samples were embedded in a Leica tissue-freezing medium (Leica Biosystem, Code 14020108926, Guadalajara, Mexico) at −26 °C. The blocks were sectioned at 30 μm with a cryostat (Leica Biosystem CM1950, Guadalajara, Mexico) with low-profile blades (Thermo Scientific, 1407060, Thermo Fisher Scientific, NJ, USA). The samples were stained with calcofluor white (18909-1000 ML-F Fluka Analytical Sigma-Aldrich, St. Louis, MO, USA) for 1 h. The images were obtained using a confocal laser scanning microscope (Olympus, FV1000 SW, Tokyo, Japan) and the FV10 ASW 3.1 viewer software. The calcofluor white signal was detected using the excitation wavelength of 380 nm; the emission wavelength was 475 nm. Sections of 30 μm were collected on glass slides. Images were taken with a microscope Axio Lab.A1 MAT HAL 50 (AXIOPLAN 490951-0002-000) and the Axiovision SE 64 Rel 4.8 viewer software.

### 4.11. Immunolocalization Assays

Immunofluorescence was assessed with modifications of the protocols previously described [2,39]. In this method, we eliminated the use of paraffin and replaced it with the Leica tissue-freezing medium (Leica Biosystem, Code 14020108926, Buffalo Grove, IL, USA) to embed the tissues. In addition, we did not use the sodium citrate buffer, Tween, and we skipped the heating step of the slides. In short, the slides with sample tissue (previously rinsed with 0.1% poly-l-lysine in H_2_O) were washed three times with sterile distilled water to remove excess Leica tissue-freezing medium, and then washed three time with the PB buffer, pH 7.2 (adjusted with NaOH 1 N). Sections were blocked with 3% bovine serum albumin (BSA, Sigma, A2153, St. Louis, MO, USA) in PB for 1 h at 4 °C. After three rinses with PB, sections were incubated overnight with anti-IAA mouse monoclonal antibody (Sigma, A0855, St. Louis, MO, USA) diluted 1:100 in 1% BSA in PB buffer. After three rinses with PB buffer, sections were incubated for 3 h in darkness with Alexa Fluor 488-labeled anti-mouse IgG antibody (Invitrogen, A-11,001, Carlsbad, CA, USA) diluted 1:100 in PB. After three washes with PB buffer, the tissue sections were treated with 10 μL of Vectashield mounting medium and DAPI to stain the DNA (Vector Laboratories, H-1200, Burlingame, CA, USA) and stored in the dark for 1 h at 4 °C. The images were obtained using a confocal laser scanning microscope (Olympus, FV1000 SW, Tokyo, Japan) and the FV10 ASW 3.1 viewer software. The IAA signal was detected using an excitation wavelength of 488 nm; the emission wavelength was 520 nm. The DAPI staining signal was detected using the excitation wavelength of 405 nm; the emission wavelength was 461 nm. The immunolocalization assay experiments were performed independently three times.

### 4.12. Controls of IAA Immunolocalization

Negative controls were performed by replacing the antibody first by PB buffer. The anti-IAA mouse antibody was incubated with a solution of 5 mg mL^−1^ synthetic IAA at a 1:2 (*v*/*v*) ratio at 4 °C overnight; the pre-blocked antibody solution was used as the primary antibody for immunofluorescence, following the same protocol and conditions as described above.

### 4.13. Statistical Analysis

The data processing to make the graphs and the statistical analyses were done with the ANOVA variance analysis program using the Origin Pro 2017 64 bit software, ver. 94E (Data Analysis and Graphing Software). Significance values were determined using the Tukey test. The differences were considered significant at *p* ≤ 0.05.

## 5. Conclusions

In this study, we demonstrated that the auxin IAA is crucial during the SE induction process in *C. canephora*. The data presented describe the expression of the *YUC* genes on Day 0 of the SE induction process. The location of free IAA on Day 0 was consistent with the analysis of gene expression, suggesting de novo biosynthesis of IAA. Additionally, exogenous application of yucasin inhibited IAA synthesis and blocked SE. In conclusion, the expression levels, the location of IAA, and the use of the yucasin inhibitor in this investigation provide valuable information for understanding IAA biosynthesis during the SE induction process in *C. canephora*.

## Figures and Tables

**Figure 1 ijms-21-04751-f001:**
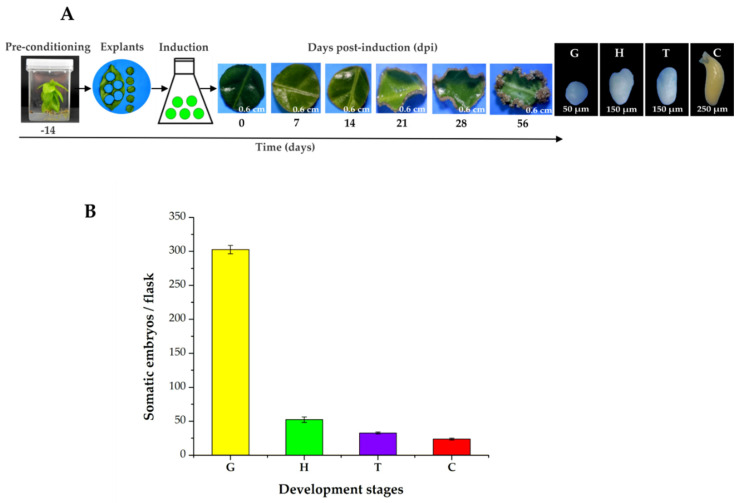
SE induction process in *C. canephora*. (**A**), On Day-14 (beginning of the pre-treatment), *C. canephora* plantlets were incubated in a pre-treatment medium (MS medium supplemented with NAA 0.54 µM and Kin 2.32 µM) for 14 days. After 14 days, explants were transferred into the induction medium (Yasuda medium supplemented with 5 µM benzyladenine) under photoperiod conditions (16/8 h) for 56 dai. (**B**), Total embryo production per flask was 402.3. The values corresponding to the different developmental stages were globular (G, 298), heart (H, 48.6), torpedo (T, 31.3), and cotyledonary (C, 24.3). The bars over the columns represent the mean value ± standard error of three independent experiments.

**Figure 2 ijms-21-04751-f002:**
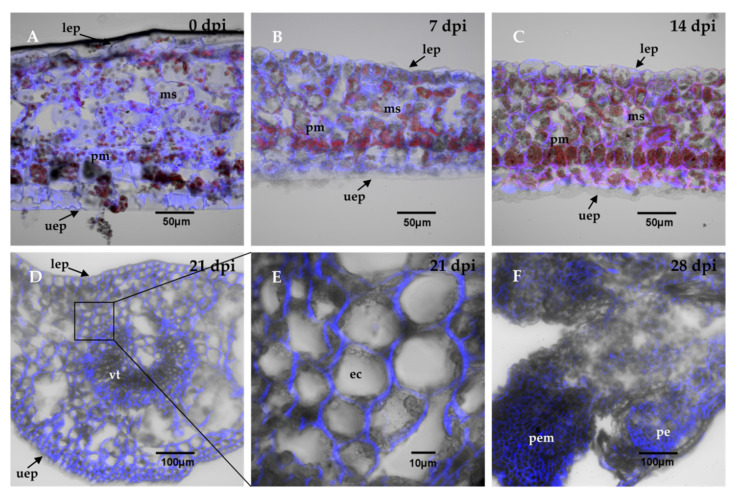
Histological analysis during the SE induction process in *C. canephora*. The leaf explants were composed of parenchymal cells of the spongy mesophyll, sm, and palisade mesophyll, pm. There were no changes in the explant cell structure, as shown in panel (**A**), which corresponds to the induction day (0 day); (**B**), 7 dai, and (**C**), 14 dai. (**D**), From 21 dai, the appearance of embryogenic cells, ec, was observed near the vascular tissue, vt. (**E**), a close-up view of the explants 21 dai showed dense embryogenic cells. (**F**), 28 dai, the proembryos, pe, were formed. The cell wall was stained with calcofluor white and chlorophyll is indicated in red. Other abbreviations: proembryogenic mass: pem; upper epidermis: uep; lower epidermis: lep. 0, 7, 14, 21, 28 dai of SE 30 µm cross-sections.

**Figure 3 ijms-21-04751-f003:**
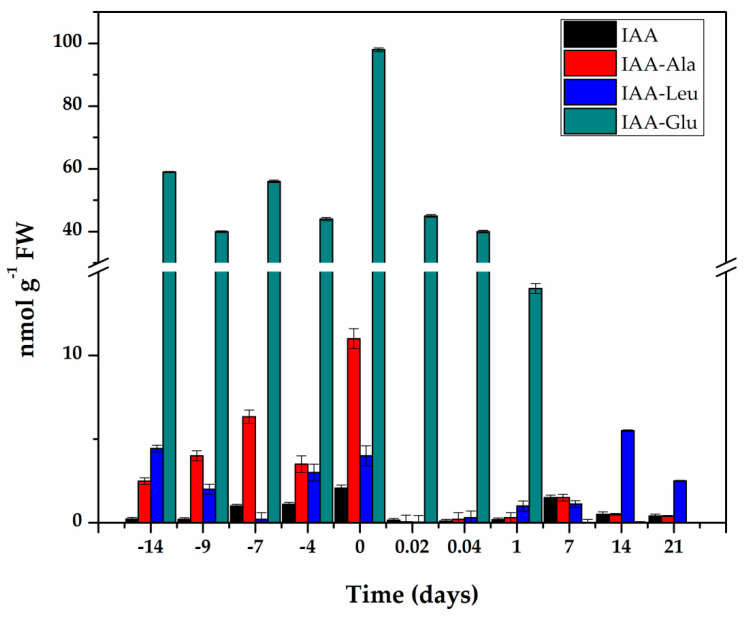
Content of IAA and its conjugates during pre-treatment and induction of SE in *C. canephora*. Samples of 100 mg of leaf tissue were collected (Days−14, −9, −4, and 0). Samples were also collected at 0.02, 0.04, 1, 7, 14, and 21 days after SE induction. Samples were analyzed as described in Materials and methods. All analyses were carried out with three biological replicates from at least two different experiments. The bars represent the standard error (*n* = 3).

**Figure 4 ijms-21-04751-f004:**
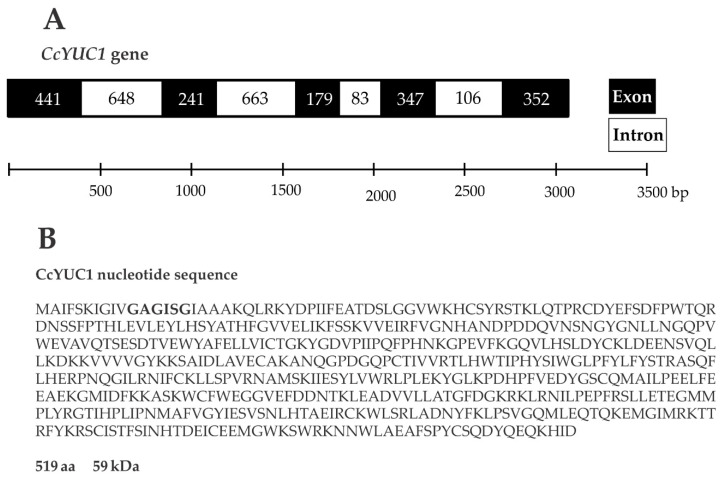
Gene structure and protein sequence of *YUC1* in *C. canephora*. (**A**), The length of the *YUC1* gene is 3060 base pairs, and the structure consists of five exons and four introns. (**B**), The *YUC1* gene coding sequence produces a 519 amino acid protein with a molecular mass of 59 kDa. Bold letters indicate the FAD binding motif.

**Figure 5 ijms-21-04751-f005:**
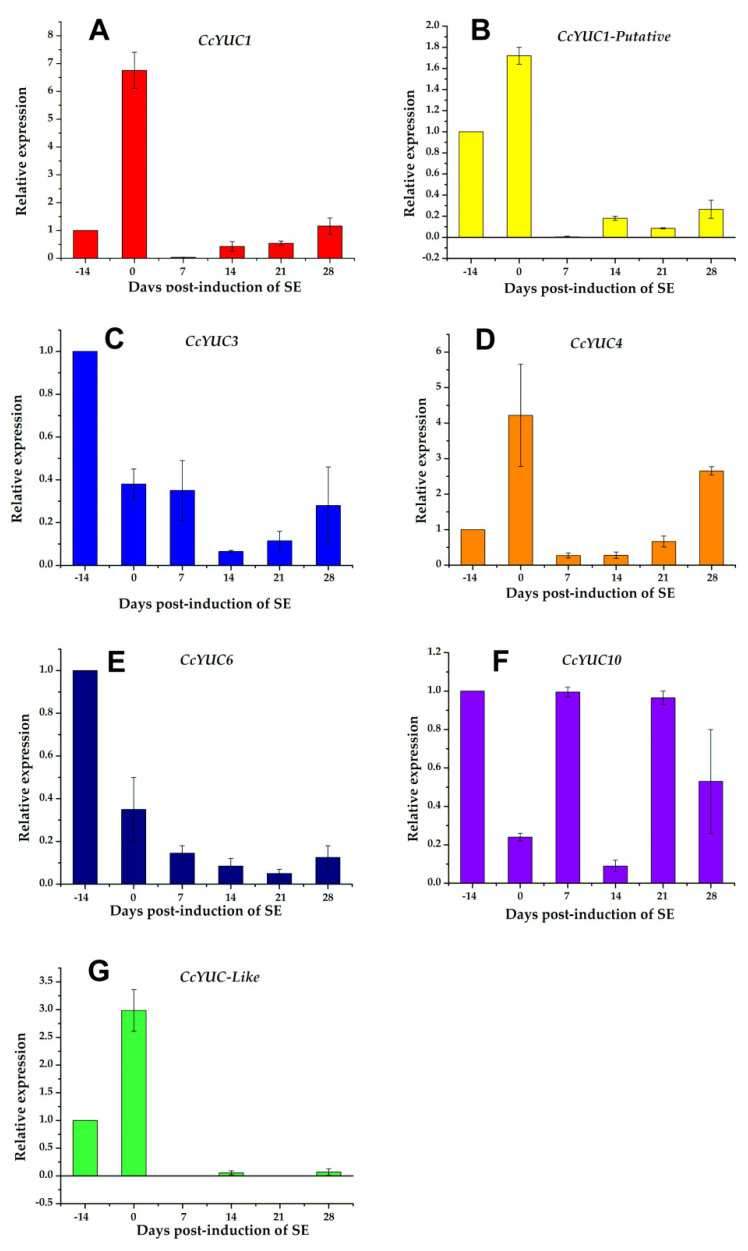
Expression levels of individual *CcYUC* genes during the induction of SE in *C. canephora*. (**A**), *CcYUC1*; (**B**), *CcYUC1*-putative; (**C**), *CcYUC3*; (**D**), *CcYUC4*; (**E**), *CcYUC6*; (**F**), *CcYUC10* and (**G**) *CcYUC*-like. Actin was used as an internal qRT-PCR reference. The bars over the columns represent the mean value ± standard error of three independent experiments.

**Figure 6 ijms-21-04751-f006:**
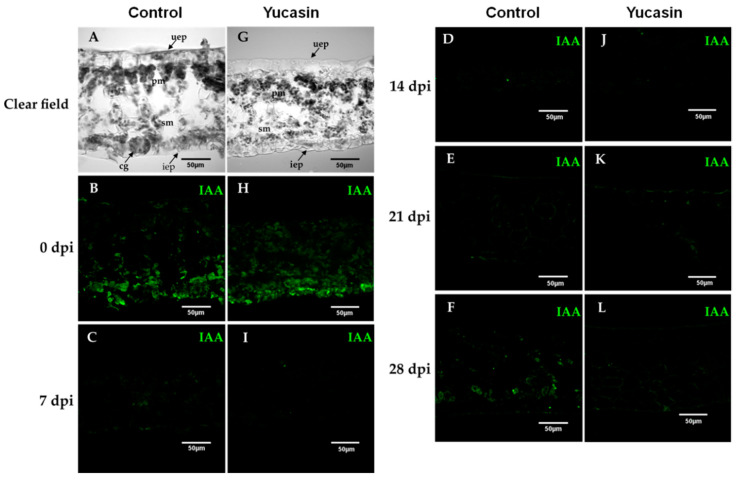
Free IAA immunolocalization during the SE induction process in *C. canephora*. Confocal images of longitudinal sections of leaf explants 0, 7, 14, 21, and 28 dai of SE without (**A**–**F**), and with yucasin (**G**–**L**). (**A**,**G**), Transmitted light differential interference images of a longitudinal sections of leaves during the induction of SE. (**B**,**H**), (0 days); (**C**,**I**), (7 dai); (**D**,**J**), (14 dai); (**E**,**K**), (21 dai); (**F**,**L**), (28 dai). IAA was visualized with the Alexa 488 chromophore bonded to the antibody that recognizes the antibody-IAA (green). Upper epidermis: uep; lower epidermis: iep; spongy mesophyll: sm; palisade mesophyll: pm.

**Figure 7 ijms-21-04751-f007:**
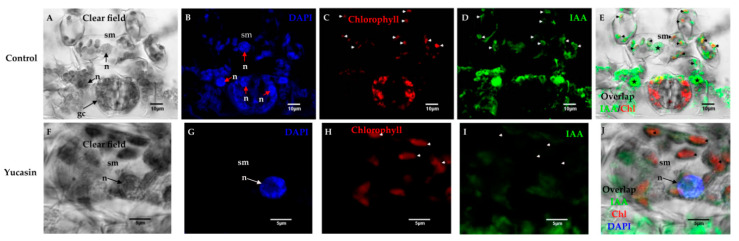
Free IAA immunolocalization on Day 0 of the SE induction process in *C. canephora*. Confocal microscopy images of cross-sections of leaf explants 0 dai of SE. (**A**), clear field; (**B**), nuclei staining with blue DAPI; (**C**), red chlorophyll signal; (**D**), free IAA signal in green; and (**E**) overlapping of the free IAA and chlorophyll signals. Note that there was IAA in the chloroplasts (short white arrows) and nucleoplasm (bold stars) of the spongy mesophyll (sm) cells; (**F**), light field; (**G**), nuclei staining with DAPI in blue; (**H**), chlorophyll signal in red; (**I**), free IAA signal in green color; (**J**), overlapping of IAA signal, chlorophyll, and DAPI on Day 0 of SE induction. IAA was not in the chloroplasts or the nucleoplasm of the spongy mesophyll cells, but in the cytosol. Mesophyll cells: sm; nucleus: n.

**Figure 8 ijms-21-04751-f008:**
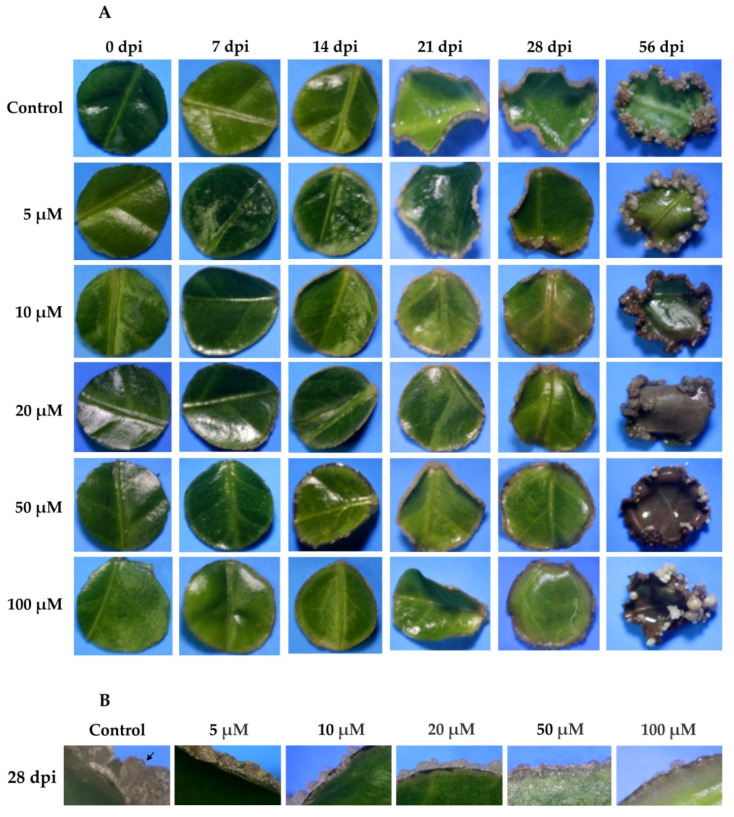
Effect of yucasin during the induction of SE in *C. canephora*. (**A**)**,** Different concentrations of yucasin were applied exogenously (5, 10, 20, 50, and 100 µM) to the pre-treatment medium. The effect of yucasin was documented every seven days until 28 dai, and then at 56 dai of SE. (**B**), A comparison of the effect of yucasin 28 dai for all the yucasin concentrations. Note the abundance of proembryogenic mass on the control, as well as the presence of proembryos (black arrow).

**Figure 9 ijms-21-04751-f009:**
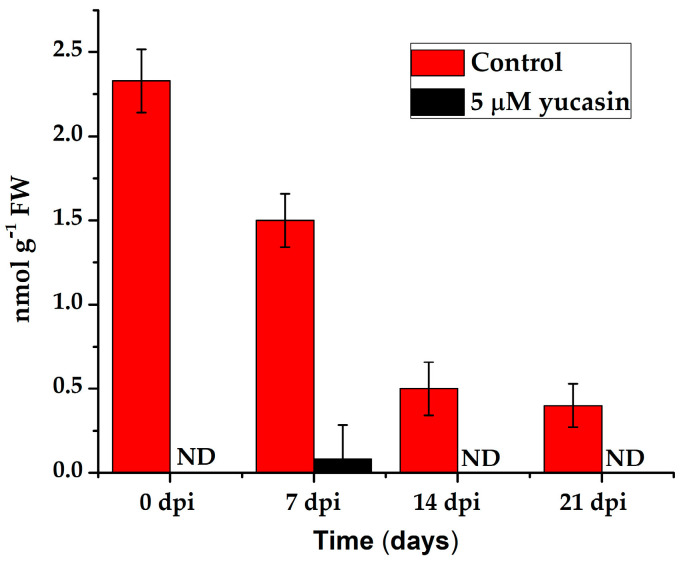
Quantification of the IAA endogenous content in leaf explants of *C. canephora* treated with 5 µM yucasin. Samples were collected at 0, 7, 14, and 21 days after SE induction. The bars over the columns represent the mean value ± standard error of three independent experiments. ND: not detected.

**Figure 10 ijms-21-04751-f010:**
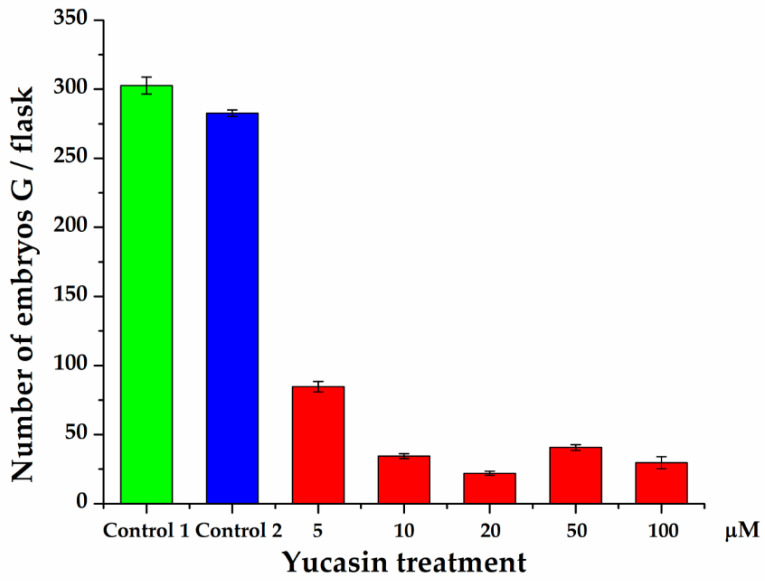
Effect of yucasin on the production of somatic embryos during the induction process in *C. canephora*. Control 1 is the treatment without any addition to the induction medium. In Control 2, DMSO was added to the semisolid pre-treatment medium. Only globular-stage embryos were formed after 56 days, so the comparison was limited to the number of globular embryos formed throughout SE induction. The data are the results of three independent biological experiments; the bars represent the standard error.

**Figure 11 ijms-21-04751-f011:**
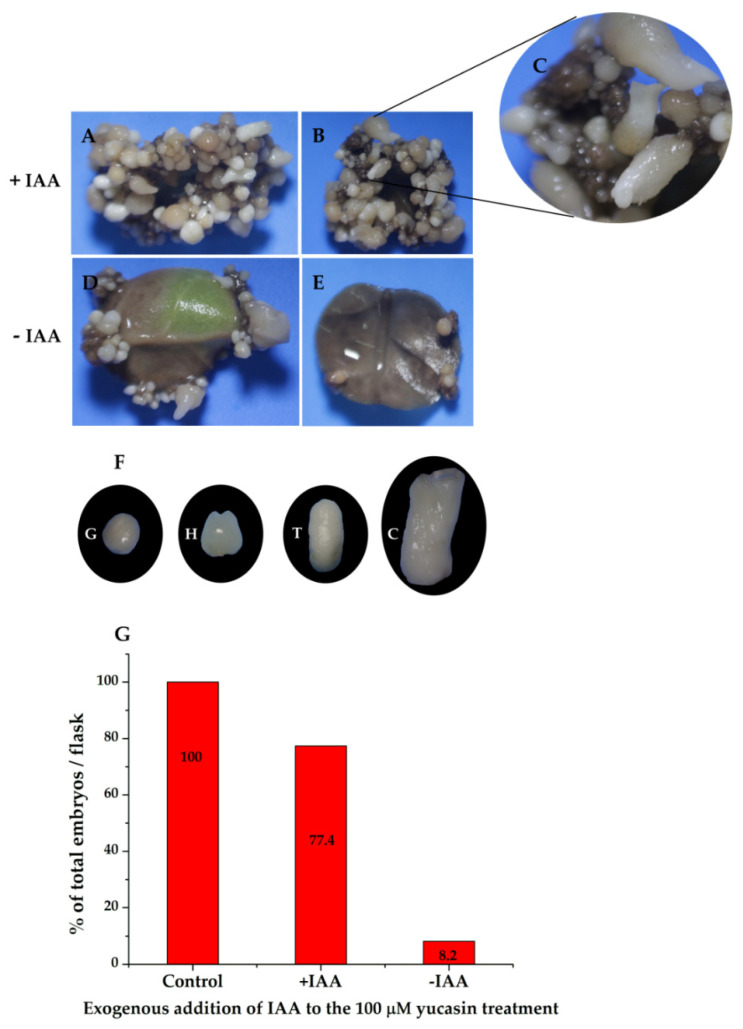
The exogenous addition of IAA restored somatic embryogenesis in explants previously treated with 100 µM yucasin. (**A**,**B**), Restoration of somatic embryogenesis by exogenous application of IAA to explants treated with 100 µM yucasin. (**C**), Close-up shows the presence of somatic embryos at different stages of development. (**D**,**E**), Explants in the presence of 100 µM yucasin. (**F**), Different development stages of somatic embryos after four weeks in the presence of exogenous IAA. (**G**), The total percentage of embryos formed in flasks with and without IAA.

**Table 1 ijms-21-04751-t001:** Total radioactivity present in each sample of *C. canephora* plantlets analyzed.

Sample		Days			
		−9	−7	−4	0
Leaf extracts	IAA	154	174	274	1093
	Trp	0	0	34	40
Leaves in medium		287	764	755	1131
Medium		2,306,388	1,845,858	1,537,818	2,283,948
Stem		-	-	-	5860
Root		-	-	-	-
Total counts per minute		2,306,829	1,846,796	1,538,848	2,292,033
Initial total cpm = 2,679,807					

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
