# Peer review of "YUCCA-Mediated Biosynthesis of the Auxin IAA Is Required during the Somatic Embryogenic Induction Process in Coffea canephora"

_ijms, 2020, doi:10.3390/ijms21134751_

Round 1

Reviewer 1 Report

The authors of the manuscript, “YUCCA-mediated biosynthesis of the auxin IAA is required during the somatic embryogenic induction process in Coffea canephora” confirmed the importance of IAA on somatic embryogenesis in C. canephora. The manuscript provides valuable information for biosynthesis and the localization of IAA during the somatic embryogenesis process in C. canephora. The results reported are encouraging. 

Some of the specific comments/suggestions are as follows:

L60: YUC expand the abbreviation

L70: Arabidopsis (Scientific name needs to be italicized)

L95: Please indicate the culture media name and plant growth regulators concentration used for pre and post SE induction.

L99: mean value ± standard error. SE- refers somatic embryogenesis

L122: proembryos instead of proembryons (Please correct it throughout the manuscript)

L130: proembryos

L146: aspartic acid (Asp), glutamic acid (Glu), alanine (Ala), and leucine (Leu).

L183: Please expand “Try”

L208: CcYUC (italic). Gene's name needs to be italicized. Please check all genes throughout the manuscript (There are several typo).

L219: induction of ES? (SE)

L238: at 28 dai

L250: IgG expand

L256: lower epidermis, iep; instead of “lower epidermis, lep;”

L290: palisade mesophyll, pm (Please indicate in Figure 7)

L313: proembryos

L327: mean value ± standard error.

L359: Please indicate control 1 and control 2 in the figure caption.

L378: Figure 11. Please move the caption to the corresponding figure.

L400: YUC

L421: 50 and 100

L427: PIN (pin-formed) and ABCB (ATP-binding cassette protein subfamily B)

Figure S1, S2, S3, S4, and S5: fragmentation pattern (please correct the typo)

Figure S6: autofluorescence instead of auto-fluorescence.

Author Response

Reviewer 1.

We followed all the specific comments/suggestions made by the reviewer.

As many of the comments were editorial aspects or typographical errors, we did not detail them (all these changes can be followed in the tracking version of the manuscript). However, we do for those who required further elaboration.

L95: Please indicate the culture media name and plant growth regulators concentration used for pre and post SE induction.

Answer.

L: 95. We incorporate the name of the culture medium, and the concentrations of the growth regulators used.

L290: palisade mesophyll, pm (Please indicate in Figure 7)

Answer.

L: 245. We apologize for the mistake. Palisade mesophyll, pm, should not have been included in the figure caption as it was not identified in the figures.

L359: Please indicate control 1 and control 2 in the figure caption. Figure 10

Answer.

L: 334. The caption of figure 10 now includes the description of how controls 1 and 2 were made.

L427: PIN (pin-formed) and ABCB (ATP-binding cassette protein subfamily B)

Answer.

L 408-409. We have incorporated full names for abbreviations.

Figure S1, S2, S3, S4, and S5: fragmentation pattern (please correct the typo)

Answer.

We have corrected the spelling error.

Reviewer 2 Report

The manuscript considers the role of indole-3-acetic acid (IAA) biosynthesis for induction of somatic embryogenesis in Coffea canephora. It was demonstrated that de novo IAA formation was necessary for somatic embryogenesis, and that embryogenesis was substantially inhibited in the presence of yucasin (specific inhibitor of YUCCA – one of the key enzymes of tryptophan-dependent auxin biosynthesis pathway).

The results described in the paper are interesting and potentially valuable. This is a resubmission, and I see that the authors made a serious work and considerably improved the manuscript. There are still some minor problems with English, but they don't hamper understanding.

I have only one specific comment:

Usually the papers are written (in particular, results are described) in the past tense. It is also the case for this manuscript, but from time to time several sentences are written in the present tense (e.g.: P. 6, Lines 193-194; P. 14, Lines 321-324). This should be corrected.

Author Response

Reviewer 2

Lines 193-194; Lines 321-324.

Answer.

L: 138. L 307-310: We have corrected the verb tense in the indicated paragraphs, and we have revised the manuscript to avoid the use of the present tense.

This manuscript is a resubmission of an earlier submission. The following is a list of the peer review reports and author responses from that submission.

Round 1

Reviewer 1 Report

The manuscript considers the role of indole-3-acetic acid (IAA) biosynthesis for induction of somatic embryogenesis in Coffea canephora. It was demonstrated that de novo IAA formation was necessary for somatic embryogenesis, and that embryogenesis was substantially inhibited in the presence of yucasin (specific inhibitor of YUCCA – one of the key enzymes of tryptophan-dependent auxin biosynthesis pathway). Generally, the results described in the paper are interesting and potentially valuable.

The manuscript needs some minor correction of English and style to make the text clearer and more concise, there are also some repetitions and misprints in the text.

Specific comments:

Page 1, Lines 13-14 – It sounds like YUCCA is the only enzyme involved in IAA biosynthesis pathway, which is not the case. Please, rephrase to make it clear that it is just one of the key enzymes. Page 1, Line 16 – It is unclear from the Abstract, what kind of “pre-treatment” you mean. Generally, this term, which is used everywhere in the paper, is not clear enough. Please, specify, what kind of pre-treatment you mean (at least in the Abstract and in the text when the term is used for the first time). Sometimes the term “preconditioning” is also used – is it the same as “pre-treatment”? Please, make it all clearer. Page 1, Line 16 – It must be “qRT-PCR”, not “qTR-PCR”. Page 1, Lines 35-36 – “…the factors involved in SE induction [6] and determine how…” Page 2, Lines 42-43 – “…culture medium, exogenous and endogenous growth regulators and nitrogen …” Page 2, Line 45 – “…growth regulators, gene expression and…” Page 2, Line 62 – Zea mays should be in Italic (Zea mays). Page 2, Line 76 – “Apparently, de novo IAA biosynthesis…” Page 2, Line 79 – “…involved in SE induction process…”; Line 81 – “…biosynthesis of IAA” Figure 1 – “Days post-induction (dpi)” in the text are named “days after induction (dai)” (Line 103) – please, choose one term and use it everywhere (check all text and figures). Page 3, Line 103 – “…the number of somatic embryos was counted.” Page 7, Line 191 – It must be “qRT-PCR”, not “qTR-PCR”. Figures 9-10 – It looks like the legend of Fig. 9 should belong to Fig. 10 and vice versa. Please, correct this. Page 13, Line 327 – “…SE is a complex process and is strictly regulated…” Page 15, Line 348 – “Consistent with this hypothesis…” Page 15, Line 369 – “…exogenous addition of IAA…” It is supposed that plants can synthesize IAA not only from Trp, but also via tryptophan-independent pathway (e.g.: https://doi.org/10.1073/pnas.1503998112; https://doi.org/10.1007/s004250000338). I think, you should consider this possibility when you discuss your data. Page 17, Lines 442-444 – Are you sure, that you first analyzed the samples with HPLC and then filtered it through a Millipore filter? Maybe, filtration was before the analysis? Page 17, Lines 458-478 – What is the difference between these two paragraphs? If it exists, please, make it clearer. Page 17, Lines 482-485 – It’s written twice, that DMSO was used as a control. Please, correct. Page 19, Line 568 (Abbreviations) – should be “kinetin” and “naphthaleneacetic acid”

Author Response

February 10th, 2020.

Mr.Jerry Wang

Assistant Editor

IJMS

Dear Mr. Wang,

We thank the reviewers for their comments and the opportunity to resubmit a revised version of our manuscript. Please find enclosed our revised manuscript, entitled “YUCCA-mediated biosynthesis of the auxin IAA is required during the somatic embryogenic induction process in Coffea canephora” by Uc-Chuc et al. We have taken the suggestions very seriously in our revision of the manuscript. The English of the manuscript was revised by Emily Wortman-Wunder, Assistant professor of English and scientific writing at the University of Colorado Denver. Please find below a detailed summary of the actions we have taken to address the reviewers’ comments.

Sincerely,

Víctor M. Loyola-Vargas, Ph.D.

Reviewer 1

Page 1, Lines 13-14 – It sounds like YUCCA is the only enzyme involved in IAA biosynthesis pathway, which is not the case. Please, rephrase to make it clear that it is just one of the key enzymes.

You are correct. We rephrased the paragraph. Lines 13-16.

Page 1, Line 16 – It is unclear from the Abstract, what kind of “pre-treatment” you mean. Generally, this term, which is used everywhere in the paper, is not clear enough. Please, specify, what kind of pre-treatment you mean (at least in the Abstract and in the text when the term is used for the first time). Sometimes the term “preconditioning” is also used – is it the same as “pre-treatment”? Please, make it all clearer.

We clarify what the pre-treatment consists of. Lines 19-20.

We also correct the term “preconditioning” to “pre-treatment through the manuscript.

Page 1, Line 16 – It must be “qRT-PCR”, not “qTR-PCR”. Page 1, Lines 35-36 – “…the factors involved in SE induction [6] and determine how…”

Thank you for your observation and we apologize for the mistake. We made the correction.

Page 2, Lines 42-43 – “…culture medium, exogenous and endogenous growth regulators and nitrogen …”

We made the correction.

Page 2, Line 45 – “…growth regulators, gene expression and…”

We made the correction.

 Page 2, Line 62 – Zea mays should be in Italic (Zea mays).

We made the correction.

Page 2, Line 76 – “Apparently, de novo IAA biosynthesis…”

We made the correction.

Page 2, Line 79 – “…involved in SE induction process…”;

We made the correction.

Page 2 Line 81 – “…biosynthesis of IAA” Figure 1 – “Days post-induction (dpi)” in the text are named “days after induction (dai)” (Line 103) – please, choose one term and use it everywhere (check all text and figures).

Thank you for your observation and we apologize for the mistake. We made the correction.

Page 3, Line 103 – “…the number of somatic embryos was counted.”

We made the correction.

Page 7, Line 191 – It must be “qRT-PCR”, not “qTR-PCR”. Figures 9-10 – It looks like the legend of Fig. 9 should belong to Fig. 10 and vice versa. Please, correct this.

Thank you for your observation and we apologize for the mistake. We made the correction.

Page 13, Line 327 – “…SE is a complex process and is strictly regulated…”

We made the correction.

Page 15, Line 348 – “Consistent with this hypothesis…”

We made the correction.

Page 15, Line 369 – “…exogenous addition of IAA…” It is supposed that plants can synthesize IAA not only from Trp, but also via tryptophan-independent pathway (e.g.: https://doi.org/10.1073/pnas.1503998112; https://doi.org/10.1007/s004250000338). I think, you should consider this possibility when you discuss your data.

Thanks for your suggestion. We wrote a small paragraph to introduce a comment in relation with the fact that there are five biosynthetic pathways for the biosynthesis of IAA, one of them independent of tryptophan.

Page 17, Lines 442-444 – Are you sure, that you first analyzed the samples with HPLC and then filtered it through a Millipore filter? Maybe, filtration was before the analysis?

Thank you for your observation and we apologize for the mistake. We made the correction.

Page 17, Lines 458-478 – What is the difference between these two paragraphs? If it exists, please, make it clearer.

Thank you for your observation and we apologize for the repetition. We eliminate the repetition.

Page 17, Lines 482-485 – It’s written twice, that DMSO was used as a control. Please, correct.

Thank you for your observation and we apologize for the repetition. We eliminate the repetition.

Page 19, Line 568 (Abbreviations) – should be “kinetin” and “naphthaleneacetic acid”

We made the correction.

Reviewer 2 Report

The manuscript reports the importance of IAA during SE in C. canephora.

Please correct the following issues:

L 55: Tryptophan Aminotransferase of Arabidopsis (TAA) and YUC

L 62: Zea mays

L 90: A) On day

L 92: B) Total

L 94: Cotyledonary correct it throughout the manuscript e.g. L 315

Figure 2 A- F dai

Please improve the quality of the photographs also indicate B and C

L 173: Delete “(TRYPTOPHAN AMINOTRANSFERASE OF ARABIDOPSIS/YUCCA) “

L 194: Bold “red” letters?

L 432: -14 missing!

L 453: Please provide the details of standards (suppliers) and provide the chromatograms of HPLC analysis of standards and samples as supplementary file.

L 497: (complementary DNA)

L 500: Coffee “Genome” Hub page

Author Response

February 10th, 2020.

Mr.Jerry Wang

Assistant Editor

IJMS

Dear Mr. Wang,

We thank the reviewers for their comments and the opportunity to resubmit a revised version of our manuscript. Please find enclosed our revised manuscript, entitled “YUCCA-mediated biosynthesis of the auxin IAA is required during the somatic embryogenic induction process in Coffea canephora” by Uc-Chuc et al. We have taken the suggestions very seriously in our revision of the manuscript. The English of the manuscript was revised by Emily Wortman-Wunder, Assistant professor of English and scientific writing at the University of Colorado Denver. Please find below a detailed summary of the actions we have taken to address the reviewers’ comments.

Sincerely,

Víctor M. Loyola-Vargas, Ph.D.

Review 2:

L 55: Tryptophan Aminotransferase of Arabidopsis (TAA) and YUC

We made the correction.

L 62: Zea mays

Thank you for your observation and we apologize for the mistake. We made the correction.

L 90: A) On day. L 92: B) Total

Thank you for your observation and we apologize for the mistake. We made the correction of both letters.

L 94: Cotyledonary correct it throughout the manuscript e.g. L 315

Thank you for your observation and we apologize for the mistake. We made the correction.

Figure 2 A- F dai

Please improve the quality of the photographs also indicate B and C

We changed the figure and complete the legend of the figure 2.

L 173: Delete “(TRYPTOPHAN AMINOTRANSFERASE OF ARABIDOPSIS/YUCCA) “

We made the correction.

L 194: Bold “red” letters?

We eliminated the “red” word.

L 432: -14 missing!

We introduced the missing data

L 453: Please provide the details of standards (suppliers) and provide the chromatograms of HPLC analysis of standards and samples as supplementary file.

Thank you for your observation. All the suppliers and catalog numbers are provided next to the name of each chemical. We include the chromatogram for the standards and two chromatograms of the samples as representative of all chromatograms obtained.

L 497: (complementary DNA)

We made the correction.

L 500: Coffee “Genome” Hub page

We made the correction.